# Improving the Detection, Assessment, Management and Prevention of Delirium in Hospices (the DAMPen-D study): protocol for a co-design and feasibility study of a flexible and scalable implementation strategy to deliver guideline-adherent delirium care

Mark Pearson [1] , Gillian Jackson,[1] Catriona Jackson [2], Jason Boland [1], Imogen Featherstone,[3] Chao Huang,[1] Margaret Ogden,[4] Kathryn Sartain,[1,5] Najma Siddiqi [6], Maureen Twiddy [1], Miriam Johnson [1]

For numbered affiliations see end of article.

**Correspondence to**
Dr Mark Pearson;
Mark.Pearson@hyms.ac.uk

## ABSTRACT

**Introduction** Delirium is a complex condition in which altered mental state and cognition causes severe distress and poor clinical outcomes for patients and families, anxiety and stress for the health professionals and support staff providing care, and higher care costs. Hospice patients are at high risk of developing delirium, but there is significant variation in care delivery. The primary objective of this study is to demonstrate the feasibility of an implementation strategy (designed to help deliver good practice delirium guidelines), participant recruitment and data collection.

**Methods and analysis** Three work packages in three hospices in the UK with public involvement in codesign, study management and stakeholder groups: (1) experience-based codesign to adapt an existing theoretically-informed implementation strategy (Creating Learning Environments for Compassionate Care (CLECC)) to implement delirium guidelines in hospices; (2) feasibility study to explore ability to collect demographic, diagnostic and delirium management data from clinical records (n=300), explanatory process data (number of staff engaged in CLECC activities and reasons for non-engagement) and cost data (staff and volunteer hours and pay-grades engaged in implementation activities) and (3) realist process evaluation to assess the acceptability and flexibility of the implementation strategy (preimplementation and postimplementation surveys with hospice staff and management, n=30 at each time point; interviews with hospice staff and management, n=15). Descriptive statistics, rapid thematic analysis and a realist logic of analysis will be used be used to analyse quantitative and qualitative data, as appropriate.

**Ethics and dissemination** Ethical approval obtained: Hull York Medical School Ethics Committee (Ref 21/23), Health Research Authority Research Ethics Committee Wales REC7 (Ref 21/WA/0180) and Health Research Authority Confidentiality Advisory Group (Ref 21/CAG/0071).

## STRENGTHS AND LIMITATIONS OF THIS STUDY

⇒ Innovative collaborative adaptation of a theoretically informed implementation strategy (Creating Learning Environments for Compassionate Care, CLECC) to deliver guideline-adherent delirium care in hospices (CLECC-Pal), including evaluation of feasibility and acceptability of an implementation strategy before testing at scale.

⇒ Research waste minimised and patient/carer burden eliminated through use of existing patient outcome and process data.

⇒ Involvement of public members since study inception and throughout study delivery and management.

⇒ While the study hospices have diverse characteristics (locations, level of socioeconomic deprivation, forms of governance), they are all drawn from a single region of the UK.

⇒ The sample size for surveys and interviews may limit the extent to which the complexity of staff and management characteristics, views and experiences can be explored.

Written informed consent will be obtained from interview participants. A results paper will be submitted to an open access peer-reviewed journal and a lay summary shared with study site staff and stakeholders.
**Trial registration number** ISRCTN55416525.

## INTRODUCTION

Delirium is a complex condition characterised by fluctuating impairment of awareness, attention and cognition.[1] Delirium causes severe distress for patients and families,[2] anxiety and stress for the health professionals and support staff providing care,[3] poor

clinical outcomes[4 5] and higher care costs (eg, longer inpatient stays).[6 7] People nearing the end of life have a high risk of delirium,[2] with risk factors such as medication, metabolic disturbance, pain, poor sleep, infection and dehydration acting cumulatively.[8] Effective delirium care is driven by prevention where possible, timely detection and non-pharmacological management, with pharmacological interventions if appropriate.[9 10] Hospices are an important but under-researched setting for the prevention and management of delirium.

An international systematic review reported that one-third of people in adult palliative care settings had delirium on admission, with two-thirds developing delirium during the admission.[8] Across health services the health economic impact of delirium is significant. Although data are not available from palliative care settings, other estimates of health service costs from delirium show comparable costs to falls, diabetes and cardiovascular diseases.[11]

National Institute for Health and Care Excellence (NICE) Clinical Guideline 103[12] and other international guidelines[13] and standards,[14] recommend strategies for delirium assessment, prevention and management. However, this is difficult in practice, with a disconnection between improved levels of delirium knowledge and the capacity of palliative care practitioners to implement changes. A recent international qualitative systematic review identified that practical and emotional support were needed to enable staff to assess, prevent and manage delirium.[15]

A recent survey of palliative care doctors (n=335) in the UK found that 38% never used delirium guidelines and that only 13% of palliative care teams used a tool (rather than clinical judgement) to assess for delirium at first inpatient assessment, with even fewer (9%) using a tool on an ongoing basis.[16] Our survey of UK specialist palliative care units (n=220, mostly nurses)[17] found that only 10% ever used a delirium screening tool, with only 5% following NICE guidelines by screening on admission, and only 6% screening daily thereafter. The importance of delirium care has been recognised in a national survey of dying patients, with 92% rating 'being mentally aware' as 'very important' and nearly as many (89%) citing 'not being a burden on family'.[18]

Delirium detection, assessment, management and prevention is complex, depending on practical support (screening tools and clinical pathways) and communication[3 19] between family and friends, volunteers, healthcare assistants, nurses allied health professionals (AHPs), social workers, doctors, hospice managers and board members. It also takes place at some of the most sensitive and emotionally fraught times in the lives of patients and their families. Therefore, guideline implementation requires a relevant and flexible strategy based on an understanding of how adaptation for different settings can be attained while retaining effectiveness.

To address this gap in knowledge about how to implement guideline-adherent delirium care, we shall first adapt an existing theoretically-informed implementation strategy that has been tested in acute hospital wards (Creating Learning Environments for Compassionate Care (CLECC)). CLECC has been found to foster and legitimise the reflection, learning, mutual support and innovation that can enable team members to progress from knowing to doing.[20] It comprises a team study day, ward manager action learning sets, peer observations of practice, and involvement of all staff in mid-shift 'cluster discussions' and twice-weekly reflective discussions,[21] and is shown mapped to the TIDieR checklist[22] in table 1. We will then test the feasibility of a subsequent quasi-experimental study to evaluate the effect of the adapted CLECC (the intervention) on hospice staff delivery of guideline-adherent delirium care and subsequent improvement in patient outcomes (reduction in the number of delirium days, with a delirium day being one where the patient was classed as having delirium using Inouye *et al*'s chart-based instrument[23]).

### Aims and objectives

This study will address key uncertainties about the implementation of guideline-adherent delirium care in hospices by demonstrating if it is possible to:

► Coadapt an implementation strategy (CLECC) for use in hospices (work package 1, WP).
► Systematically and reliably collect data (including delirium diagnosis) from clinical records in a way that minimises burden for patients, families and staff (WP 2).
► Collect measures of staff engagement with the implementation strategy, delivery of guideline-adherent delirium care, and the costs of staff involvement (WP 2).
► Collect explanatory process data about staff use of the implementation strategy (WP 3).
► Estimate the number of hospice sites and in-patient episodes needed for the planned national quasi-experimental study.

WP 1 commenced June 2021, with WP 2 and 3 (and data collection) commencing August 2021. The study will be completed in February 2023.

### METHODS AND ANALYSIS
#### Design summary

Table 2 presents the research questions and summarises the three WPs that will enable the above aims and objectives to be met. Figure 1 shows the study timeline and how the WPs are inter-related.

### Settings

Three adult hospices in northern England (UK). Two hospices in this study are located in socio-economically deprived urban areas (one with a significant minority ethnic group population) and one hospice in an affluent rural/urban area. One hospice is run by a national charity, with the other two hospices run by independent charities.

**Table 1** CLECC[21] components mapped using Template for Intervention Description and Replication (TIDieR) checklist[22]

| Component | Why | What | Who | How | Where | When/How much | Tailoring and modifications | Fidelity |
|---|---|---|---|---|---|---|---|---|
| Study day | Prepare staff for the workplace elements of the intervention | Procedure: Introduction to CLECC Activities/discussion Questionnaires Film handouts | Appointed hospice lead clinician | Classroom based to include all hospice staff | Comfortable classroom that is geographically separate from the workplace | One day at beginning of implementation period, but may require more than one study day to ensure maximum attendance | Pending work package 1 codesign workshops | Attendance and feedback data from hospice lead clinician. |
| | | Materials: PowerPoint presentation. Record of attendance. Summary of CLECC leaflet | | | | | | |
| Action Learning sets | Real problems from own practice and devise action plan to address | Procedure: Session 1: relationships and rules Session 2: valuing staff Session 3: enhancing capacity CLECC Session 4: influencing seniors | Experienced facilitator and 4 to 8 leads of comparable position | Face to face at hospice site | At hospice site | 4×4 hours action learning sets throughout intervention period | Pending work package 1 codesign workshops | Fidelity/attendance |
| Peer review | Appreciate practice from observer perspective | Procedure: 2–3 × 1 hour observations Reflective summary Materials: Training video Poster of findings | 2 team members nominate or nominated by lead and training given. | Outside of normal role to do this activity | At hospice site | Approximately 30 min training video prior to commencing 2–3 × 1 hour observations throughout implementation | Pending work package 1 codesign workshops | Fidelity |
| Mid-shift cluster discussions | Opportunities for feedback, group problem solving and support to individual team members. | Procedure: Mid-shift 5 min discussion | All team members on shift. | Mid-way through every shift. | At hospice site | 5 min discussion mid-shift, initially instigated by lead but then to be maintained by staff | Pending work package 1 codesign workshops | Fidelity |
| Reflective discussions | To prompt personal reflections and narratives about individual experiences | Procedure: Scheduled meetings or drop in sessions with planned activities Materials: Devise a sustainability plan | All team members, including senior staff and temporary staff. | Can be scheduled time during shift or drop-in sessions. | At hospice site, in a comfortable room on or near place of care. | No of sessions dependent on the number of subjects needed to be discussed | Pending work package 1 codesign workshops | Fidelity |

CLECC, Creating Learning Environments for Compassionate Care.

**Table 2**  Overview of study design

| Work package objective | Research question | Study type | Data collection | Timepoints |
|---|---|---|---|---|
| 1. Refine CLECC-Pal implementation strategy | What are the core and adaptable components of an implementation strategy for guideline-adherent delirium care in hospices? | Experience-based codesign | Workshops | Before and during implementation |
| 2. Demonstrate feasibility of future quasi-experimental study | Is it feasible to collect sufficient outcome data (both implementation and clinical), explanatory process data, and cost data in a future effectiveness evaluative study in palliative care settings? | Feasibility study | Patient demographics and delirium diagnosis and management (clinical records) No of staff engaged in CLECC-Pal | Baseline and follow-up During implementation and follow-up |
| 3. Assess acceptability and flexibility of CLECC-Pal implementation strategy | How can a codesigned implementation strategy for guideline-adherent delirium care be operationalised with fidelity to function in different hospice inpatient settings? | Realist process evaluation | Survey Fidelity to CLECC-Pal Interviews | Baseline and follow-up Start, middle and end of 3-month period using CLECC-Pal Follow-up |

CLECC, Creating Learning Environments for Compassionate Care.

## Patient and public involvement

This study supports the involvement of patient and public involvement (PPI) in accordance with the framework for good public involvement as detailed by the UK standards for public involvement.[24] Public involvement group members contributed to study design, with one member joining the monthly Study Management Group meetings, cofacilitating workshops (WP 1) and a further member Chairing the Study Steering Committee. The study's public involvement group will meet three to four times over the duration of the study to discuss public involvement challenges in the research, the implications of emerging study findings, and the development of public-facing research outputs and the next steps in the research cycle.

## WP1: adaptation (codesign) of CLECC for guideline-adherent delirium care

An experience-based codesign group[25–27] of people with lived experience of delirium (themselves or in a family member or friend), staff and management from across the study sites and the region will meet for online workshops (maximum 3 hours duration) at months 2, 8 and 14 to adapt the CLECC strategy for use in hospices (see figure 1). The first of these codesign workshops will be held separately for public and staff to facilitate reflection within a broader public or staff 'group' and to underpin interactions between public and staff at subsequent joint workshops. The interactions in these joint codesign workshops are considered essential for participants to share their experiences, develop an appreciation of others' experiences, and open up new ways of thinking about how to meet challenges that will directly inform codesign.[28] Consistent with the INVOLVE principles for coproducing research,[29] workshops will be codeveloped with our public involvement group and cofacilitated by an experienced Public Involvement group member.

Potential public participants will be invited through existing national PPI networks to join the codesign workshops. Potential hospice staff and management participants (clinicians, volunteers, managers and board members) will be invited through existing communication channels at each site and in consultation with managers. Information will be provided for potential participants with an opportunity to discuss in more detail prior to taking part. Workshops will be scheduled to fit with existing commitments and day-to-day practice at each hospice. PPI team member (MO) will provide input into all aspects of invitations, information provision and workshop design.

We shall endeavour to maximise diversity within the workshops but acknowledge the tension between attaining diversity across every potential aspect and a maximum workable number of workshop participants of around 15. We shall keep this under review with PPI team member MO.

Central to the conduct of the workshops will be the use of 'touch points' to communicate other peoples' experiences and provide a focus to spark discussion and exploration from different perspectives.[26] Touch points are the events which significantly shape people's positive or negative experience of an event or service. It could be the sharing of a personal or professional experience of delirium care by a workshop participant, or a short film or news item about palliative care services generally or delirium specifically. These will be used to trigger discussion about the detection, assessment, prevention and management of delirium, how CLECC can be adapted for hospices and support implementation of delirium guidelines.

Table 3 provides an overview of the schedule and content of the codesign workshops.

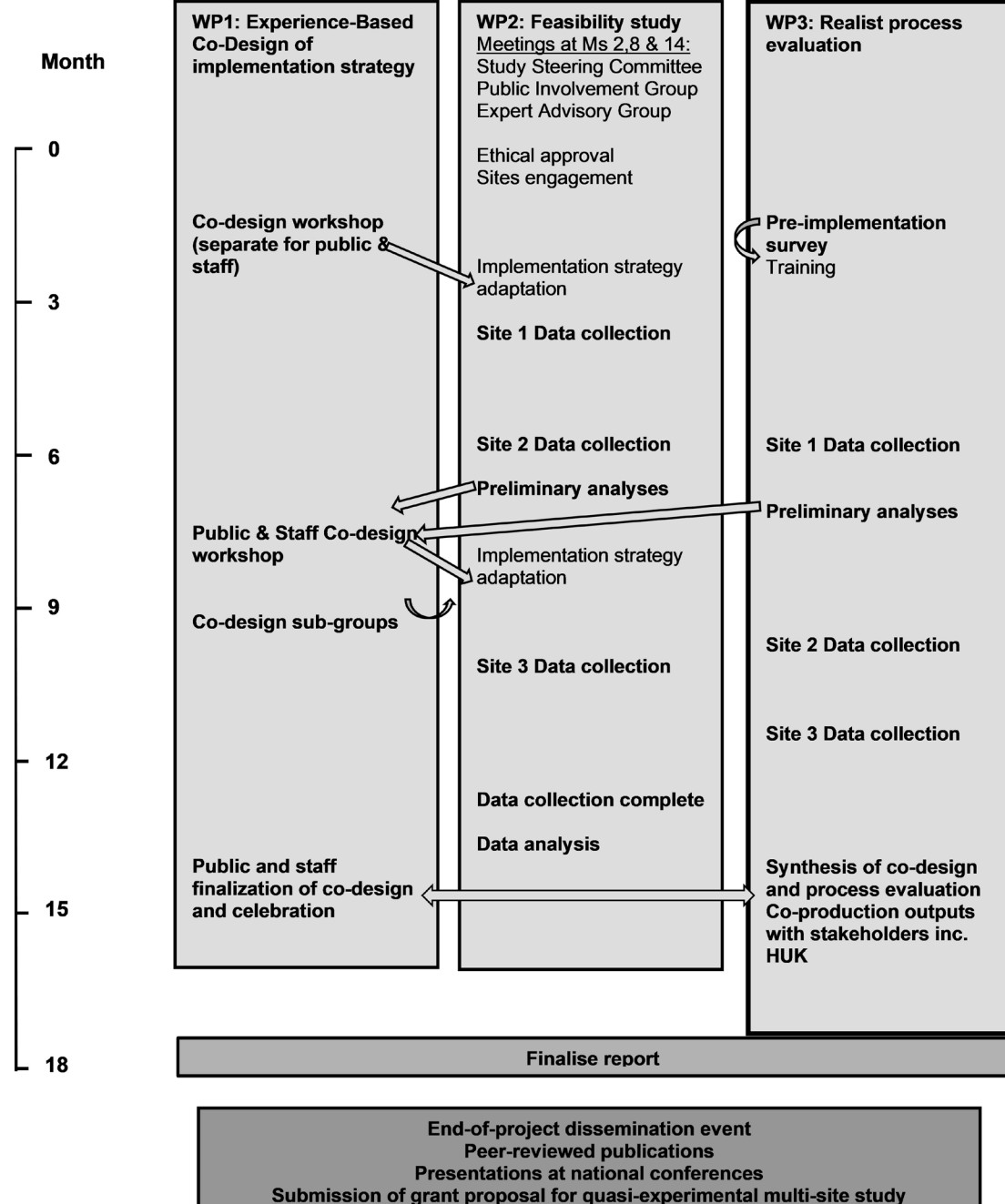

**Figure 1** Study flow chart and timeline summary. WP, work package.

## WP 2: feasibility study

Feasibility will be assessed in the following key areas:

► Patients:
– Ability to collect high quality, anonymised delirium outcome and process (extent of guideline-adherent care) data from clinical records.
– Variability of baseline delirium day measurement to calculate the sample size for a subsequent national study.

► Staff and volunteers: Number of relevant hospice staff and volunteers' participation in CLECC-Pal activities (proportion of relevant staff engaging and maintaining engagement).

► Economic: Ability to collect cost data in relation to CLECC-Pal staff activities.

The codesigned CLECC-Pal (for initial version, see table 1) will be introduced to clinical and support staff, volunteers and managers at each hospice in a study day that will include training in guideline-recommended delirium care. The study team will support the identified clinical lead to introduce and use CLECC-Pal, including action learning sets, mid-shift 'cluster discussions', twice-weekly reflective discussions and peer observations of practice, over a minimum 12-week period. The study day ethos will emphasise how hospices should take ownership of using CLECC-Pal with only modest support from the study team.

**Table 3**  Codesign workshops schedule and content

| Workshop focus | Participants | When, duration | Content |
|---|---|---|---|
| 1a. Introduction and initial refinement of CLECC-Pal | Public members | Month 2, 2 hours | ▸ Introductions<br>▸ Discussion about the principles of equitable participation<br>▸ Discussion about the codesign approach to workshops<br>▸ Introduction to the CLECC strategy and exploration of priority aspects for adaptation<br>▸ Identification of individual working groups' role in exploring and refining site-specific or issue-specific aspects of the CLECC strategy before Workshop 2<br>▸ Agreement on feedback processes outside of the workshops and focus of agenda for Workshop 2 |
| 1b. Introduction and initial refinement of CLECC-Pal | Hospice staff and volunteers | Month 2, 2 hours | As for Workshop 1a |
| 2. Refinement of CLECC-Pal | Public members, hospice staff and volunteers | Month 8, 3 hours | ▸ Feedback from individual working groups<br>▸ Discussion of emerging findings from work package 3 (realist process evaluation)<br>▸ Specification of suggested adaptations to CLECC<br>▸ Identification of further individual working groups to refine site- or issue-specific aspects of the CLECC strategy<br>▸ Agreement on focus of agenda for Workshop 3 |
| 3. Final specification of CLECC-Pal and celebration | Public members, hospice staff and volunteers | Month 14, 3 hours | ▸ Feedback from individual working groups<br>▸ Discussion of further findings from work package 3 (realist process evaluation)<br>▸ Final specification of adaptations to CLECC<br>▸ Celebration of codesign outputs |

CLECC, Creating Learning Environments for Compassionate Care.

## Data collection and analysis

### Patients

Baseline and follow-up (pre and post) clinical record data will be collected. Data will be collected through remote access to the clinical record where electronic records allow, or from the paper record. At each of the three hospices, case note collection (total n=300) will comprise:

▸ Baseline (pre): 50 consecutive patients who completed their in-patient stay immediately prior to the start of the hospice using CLECC-Pal.

▸ Follow-up (post): 50 consecutive patients completing their in-patient stay from week 4 of starting use of CLECC-Pal.

Clinical record data collected by the researcher will be anonymised at the point of extraction and include:

▸ Demographic data (baseline only): age, sex, main medical condition, ethnicity, postal code (converted to Index of Multiple Deprivation (IMD) score).

▸ Delirium diagnosis using the Inouye *et al* case note tool.[23]

▸ Delirium management: including evidence of use of delirium screening tools, risk assessments and individualised delirium management care plans.

Clinical record data will be extracted using an expanded version of the prospectively validated (74% sensitivity, 83% specificity) chart-based instrument developed by Inouye *et al* for detecting potential delirium diagnoses from clinical records.[23] The instrument (data extraction proforma, see online supplemental file 1) will enable us to assess whether case-note recorded symptoms of delirium (and therefore number of patient days with delirium) can be linked to time points during the person's admission when actions around delirium assessment, management and prevention (consistent with guidelines) did or did not take place. Our 'expanded' version of the instrument will include questions about other actions to support delirium assessment, management and prevention that may be recorded in the notes, as shown in table 4. We shall report the percentage of clinical records where information about each of these actions is recorded. Where a person experiences multiple episodes of delirium within one admission, each episode will be recorded separately and linked through the anonymised case number.

Where judgements about what to record on the proforma need to be made, justification for these will be recorded on the form. Any uncertainty about how the information in the case notes should be recorded on the proforma will be discussed with a second clinician (CJ) and justification for the final decision recorded.

The number of patient records from which it was possible to extract clinical record data longitudinally over the duration of their inpatient admission will be reported both as a simple count and as a percentage of the total number of in-patients with a diagnosis of delirium in each hospice each month.

**Table 4** Additional delirium assessment items to be derived from clinical records and means of assessing feasibility of data collection

| Delirium-related action | Assessment of feasibility |
|---|---|
| Use of richmond agitation-sedation scale and 4AT screening tools | % completed<br>% reassessments completed at appropriate timepoints |
| Medication reviews (to minimise deliriogenic medication) | % completed<br>% reassessments completed at appropriate timepoints |
| Diagnostic and Statistical Manual of Mental Disorders (DSM-V) delirium assessment | % completed<br>% reassessments completed at appropriate timepoints |
| Degree of sedation or agitation | % completed |
| Individualised delirium care plans | % completed<br>% reviewed at appropriate timepoints |
| Presence/absence of delirium | % documenting start and end of delirium episode(s)<br>% documenting delirium-free days |

Sample size: Based on our pilot work in one hospice (comparable in size to the hospices in this study) which identified a monthly occurrence of 32 inpatient episodes of delirium, retrospectively collecting clinical record data for all patients whose episode of in-patient care is completed (up to a maximum of 50 per hospice) will provide us with enough data to answer feasibility questions about data quality and enable us to capture frequent events regarding care planning. We do not propose to investigate less-frequent events such as antipsychotic use.

Analysis: Baseline demographic and clinical characteristics of the study population (age, sex, primary medical condition, ethnicity, postal code (to derive IMD)) will be presented using descriptive statistics. Mean (SD) will be reported for continuous data and raw count (number, percentage) will be reported for nominal data. The variation around baseline delirium days will be calculated to inform the sample size and number of hospices needed for the subsequent national study.

### Staff and volunteers

In consultation with operational and clinical management at each site, a hospice study lead has been identified through whom the following denominators will be established:
- ► Number of staff working on or rotating through the in-patient unit of the hospice.
- ► Number of volunteers active within the in-patient unit of the hospice.
- ► Total number of documented in-patient delirium episodes or (if total number cannot be established) number of patients with at least one case-note

diagnosis of delirium per in-patient admission in the hospice.

Level of staff engagement with CLECC-Pal during the implementation period will be assessed weekly by the hospice study lead completing a rapid report of numbers of:
- ► Staff indirectly involved in delivering delirium care who attend the team study day, action learning sets, feedback following peer observations of practice, mid-shift cluster discussions, and reflective discussions.
- ► Staff and volunteers who do not engage with CLECC-Pal.
- ► Staff and volunteers who decrease or stop their engagement with CLECC-Pal.
- ► Peer observations of practice achieved.
- ► People approached, reported by professional group and role, who agree to participate in using CLECC-Pal.

The rapid report will also record reasons for:
- ► Staff and volunteers' non-engagement or dropout.
- ► Modifications made in the use of CLECC-Pal.

Quantitative data will be analysed descriptively using radar plots. Qualitative data will be rapidly analysed deductively using a Framework approach.[30] Analyses will inform more detailed exploration in interviews (WP3) and will be shared with participating hospices to inform their ongoing use of CLECC-Pal.

### Economic

We will assess the feasibility of collecting data about the costs of using CLECC-Pal:
- ► Number of hours spent by members of staff and volunteers in CLECC-Pal activities, linked to pay-grade where possible.

### WP 3: realist process evaluation

Critiques of process evaluations have highlighted the importance of methods that can use theory to explore how contexts and mechanisms interact,[31–33] as recognised in the revised Medical Research Council framework.[34] We shall use realist evaluation[35] to capture staff and management insights into how individual-level, team-level and organisational-level contexts affect these interactions during implementation,[36] refining normalisation process theory's (NPT) propositions about the mechanisms of coherence, cognitive participation, collective action and reflexive monitoring.[37–39] Definitions of realist terms are shown in table 5. This theoretically informed understanding of how the implementation strategy functions[40] will enable us to explain how hospices may operationalise CLECC-Pal in different ways to achieve the same desired outcomes (eg, by running online learning rather than a team study day, or using self-reflection on practice rather than peer observation).

### Identification, sampling and consent
#### Surveys

All hospice staff involved in direct patient care or management, as well as those directly involved in patient care

**Table 5** Definition of realist terms used in work package 3

| Term | Definition |
|---|---|
| Context | Individual, team, organisational or other factors that enable or constrain the operation of mechanisms.[53] This includes social phenomena such as rules, norms and values, meaning that contexts are not straightforwardly analogous with settings.[54] |
| Mechanism | The interaction of a programme's resources or opportunities with individuals' or teams' reasoning.[53] |
| Outcome | The 'demi-regular' occurrences arising from particular configurations of contexts and mechanisms.[47] Consistent with the recognition in realist ontology of the dynamic and non-linear nature of open systems in the social world,[55] 'outcomes' may be better understood as semistable processes. |
| Programme theory | A middle-range theoretical explanation of how (implementation) programme activities relate to underlying theory. Even if not explicitly stated, programme theories contain ideas about how best to address challenges to achieving intended goals (including how to proactively manage these challenges)[56] |

(volunteers, support staff, board members with a hospice governance role) will be eligible. Minimum sample size of 10 at each hospice (total n=30). Eligible participants will be sent a link to the anonymous survey, for which completion online will be taken as implied consent.

### Interviews

A purposive sampling strategy at each site will draw from a sampling frame that includes all hospice staff involved in direct patient care or management, volunteers, support staff and board members at each study site. Within the constraints of an exploratory sample size (five staff and volunteers, and two members of management and/or executive board at each site; minimum total n=15), we shall endeavour to maximise variation in participant characteristics and roles, prioritising sampling that will enable comparison between those who do and do not take part. Written informed consent will be obtained. Interviews will be conducted at a time suitable for participants and may be face to face or remote, according to participant preference.

### Data collection and analysis

#### Staff and volunteers' preimplementation and postimplementation experiences (survey)

Survey using a modified and piloted normalisation measurement instrument (NoMad).[41] of staff and volunteers' perceptions and experiences of implementation, in relation to each NPT mechanism, before and after using the CLECC-Pal implementation strategy.

Quantitative Likert scale responses will be analysed descriptively using radar plots. Free-text responses will be

deductively thematically analysed using the framework of NPT mechanisms (coherence, cognitive participation, collective action and reflexive monitoring), allowing for inductive thematic analysis if responses do not fit within the framework. Thematic patterns and outliers will be identified. The analysis will also inform the structure, content and focus of the staff and volunteer interviews.

#### Staff and volunteers' postimplementation experiences (interviews)

Realist interviews are distinct from conventional qualitative semistructured interviews as they adopt a 'teacher-learner' approach. This involves presenting theory to participants so that they can communicate their own experiences and views that may refute, refine, or expand the theory.[42] In practice, the realist interviewer presents theory (context-mechanism-outcome configurations) in a form comprehensible to the participant and follows-up flexibly with further questions tailored to the participant's understanding, to ensure that the discussion enables theory-refinement rather than simply a discussion of experiences. Interviews will build on Murray et al's[43] operationalisation of NPT for the development and optimisation of interventions within trials (see table 6).

Interview topics will include, but not be limited to, experiences of CLECC-Pal's acceptability and fit, rationale for any modifications to CLECC-Pal, perceived changes in communication between those caring for patients at-risk of delirium, changes in care practices, perceptions about how CLECC-Pal is achieving (or not) the intended effects and, if appropriate, how these impacts could be sustained. Interview questions will be informed by emerging site-specific data from the codesign and feasibility WPs, as well as from the process evaluation survey. Graphical summaries of data, such as radar plots, will be used in the interviews to communicate this emerging data to participants, link to theory and to support discussion that enables implementation theory to be refined.[42 44] Views of study processes will also be sought. It is envisaged that interviews will last no longer than 30 min, but participants will be given the opportunity for a longer interview if they wish.

Interviews will be recorded and transcribed. Before commencing analysis, interview transcripts will be read and reread to allow familiarisation with the content that will enable theory-building and refinement rather than rote coding of contexts, mechanisms and outcomes (although coding of these configurations may also play an important role in theory-building and refinement). Analysis to identify contextualised explanations of how mechanisms of implementation are understood to lead to certain outcomes will be structured using the reasoning processes identified by Pawson (juxtaposition, reconciliation, adjudication, consolidation and situating[45]). We shall operationalise these reasoning processes using the analytic questions for building and refining programme theory identified by Pearson et al.[46]

WP 3 methods and findings will be reported consistent with the RAMESES reporting standards.[47]

**Table 6** Normalisation process theory 'contribution' mechanisms and their relationship to data collection in interviews

| Mechanism | Definition[37] | Theoretical propositions[38] | Potential interview questions[43] |
|---|---|---|---|
| 1. Coherence | Agents attribute meaning to a complex intervention and make sense of its possibilities within their field of agency. They frame how participants make sense of, and specify, their involvement in a complex intervention. | 1.1 Embedding is dependent on work that defines and organises a practice as a cognitive and behavioural ensemble. 1.2 Embedding work is shaped by factors that promote or inhibit actors' apprehension of a practice as meaningful. 1.3 The production and reproduction of coherence in a practice requires that actors collectively invest meaning in it. | Is CLECC-Pal: ► easy to describe? ► clearly distinct from other strategies? ► have a clear purpose for all participants? Do participants have a shared sense of purpose? What benefits will the intervention bring and to whom? Are these benefits likely to be valued by potential participants? Will CLECC-Pal fit with the overall goals and activity of the organisation? |
| 2. Cognitive Participation | Agents legitimise and enrol themselves and others into a complex intervention. They frame how participants become members of a specific community of practice. | 2.1 Embedding is dependent on work that defines and organises the actors implicated in a practice. 2.2 Embedding work is shaped by factors that promote or inhibit actors' participation. 2.3 The production and reproduction of a practice requires that actors collectively invest commitment in it. | Are target user groups likely to think that CLECC-Pal is a good idea? Will they see the point of CLECC-Pal? |
| 3. Collective Action | Agents mobilise skills and resources and enact a complex intervention. They frame how participants realise and perform the intervention in practice. | 3.1 Embedding is dependent on work that defines and operationalises a practice. 3.2 Embedding work is shaped by factors that promote or inhibit actors' enacting it. 3.3 The production and reproduction of a practice requires that actors collectively invest effort in it. | How will CLECC-Pal affect the work of user groups? Will CLECC-Pal promote or impede their work? Will staff require extensive training before they can use CLECC-Pal? How compatible with existing work practices is CLECC-Pal? What impact will CLECC-Pal have on division of labour, resources, power, and responsibility between different professional groups? Will CLECC-Pal fit with the overall goals and activity of the organisation? |
| 4. Reflexive Monitoring | Agents assemble and appraise information about the effects of a complex intervention within their field of agency, and use that knowledge to reconfigure social relations and action. They frame how participants collect and use information about the effects of the intervention. | 4.1 Embedding is dependent on work that defines and organises the everyday understanding of a practice. 4.2 Embedding work is shaped by factors that promote or inhibit appraisal. 4.2 The production and reproduction of a practice requires that actors collectively invest in its understanding. | How are users likely to perceive CLECC-Pal once it has been used for a while? Is CLECC-Pal likely to be perceived as advantageous for patients or staff? Will it be clear what effects CLECC-Pal has had? Can users contribute feedback about CLECC-Pal once it is in use? Can CLECC-Pal be adapted or improved on the basis of experience? |

CLECC, Creating Learning Environments for Compassionate Care.

## ETHICS AND DISSEMINATION

Ethical approval for the study has been obtained from Hull York Medical School Ethics Committee (Ref.: 21/23), Health Research Authority Research Ethics Committee Wales REC7 (Ref.: 21/WA/0180) and Health Research Authority Confidentiality Advisory Group (Ref.: 21/CAG/0071). Confidentiality advisory group approval allows the study researcher access to the clinical records to extract data without patient consent. The study is publicised in the hospices during the data collection period and patients/representatives may opt out if they do not wish their data to be used. Written informed consent will be obtained from interview participants.

he primary objective of this study is to inform a future quasi-experimental multi-site comparative evaluation. We shall do this by demonstrating the feasibility (or otherwise) of the implementation strategy ('intervention'), participant recruitment and data collection, in

addition informing decisions about the most appropriate study design for a future multisite comparative evaluation. However, as argued by Thabane et al[48] communicating findings from feasibility studies remains critically important for ensuring that resources are not spent on either duplicating the feasibility study or funding research uninformed by the findings of a relevant feasibility study. We shall therefore prepare a full report of the study's methods and findings for the funder and submit a manuscript reporting the findings to an open access peer-reviewed journal. The study's findings will also be submitted for oral presentation at one national health services research conference and one international palliative care conference. A Plain English summary of study findings will be prepared for distribution through palliative care clinical networks (including Hospice UK) and public involvement groups.

## DISCUSSION

This study will address key uncertainties about the implementation of guideline-adherent delirium care in hospices—the feasibility of using a theoretically informed, codeveloped implementation strategy (CLECC-Pal); collecting demographic, diagnostic and delirium management data from clinical records; collecting measures of staff engagement; and collecting explanatory process data about staff use of CLECC-Pal. This will enable us to estimate the number of hospice sites and in-patient episodes needed for the planned national quasi-experimental study, for which we outline the design considerations below. The study has clear strengths in public involvement and in minimising research waste by using existing process and outcome data. There are also limitations in the study, for example, hospices are all drawn from a single region of the UK and the sample size for surveys and interviews may limit the extent to which the complexity of staff and management characteristics, views and experiences can be explored. Nevertheless, the study hospices have diverse characteristics (locations, level of socio-economic deprivation, forms of governance) and we shall purposively sample staff and management (for interviews) to maximise the range of professional and role characteristics.

We have developed this feasibility study to inform future decisions about evaluative study design that balances scientific rigour and practical considerations. In doing so, we first appraised an interrupted time series design that would enable naturalistic data collection, but considered this unrealistic as powering the study would likely require 12 months preintervention and postintervention data collection.[49] Second, we appraised a randomised stepped wedge design, but considered implementation research permutations of this design unlikely to be feasible due to the real-world setting (if using a head-to-head rollout design) or length of time required (if using a pairwise enrolment rollout design).[50]

Consistent with current thinking in implementation research for investigators to consider quasi-experimental study designs that can assess the impact of context over time,[51] we plan to work towards an evaluative study design that uses natural variation in the introduction of the implementation strategy to allow a non-randomised stepped wedge design (CLECC-Pal supported delirium care vs delirium care as usual). Our audit data indicate that this would be realistic given an annual admission rate of 192–384 in the 10–20 bedded study site hospices which have a 40%–60% incidence of delirium.

While hospices are relatively homogeneous in terms of care delivery by health professionals (eg, standardised national training programme for doctors, national standards for nursing practice), the wide referral base of hospices mean that in-patients tend to be heterogeneous in relation to type and stage of disease, ethnicity, socioeconomic status and so on. For the future evaluative study, we shall estimate the intraclass correlation coefficient using preintervention patient outcome data (delirium-free days) from the feasibility study, thus enabling a sample size calculation powered on the primary outcome for the future evaluative study.

We are mindful of a recent systematic review of feasibility studies which identified a lack of consistency in the use of terminology, a predominance of feasibility issues relating to preparation for randomised-controlled trials, and an absence of clear guidance about when 'sufficient insight about uncertainties' had been achieved for progression to an evaluation study.[52]p.10 However, we are confident in stating minimum recruitment targets for the use of CLECC-Pal (fidelity to core components) and 4AT screening tool at baseline and daily, that will be necessary for a future evaluative study to be considered feasible:

► ≥80%, proceed.
► 60%–80% with mitigating factors, proceed.
► <60% not feasible.

**Author affiliations**
[1]Hull York Medical School, University of Hull, Hull, UK
[2]St James's University Hospital, Leeds, UK
[3]Department of Health Sciences, University of York, York, UK
[4]Faculty of Social Sciences, University of Stirling, Stirling, UK
[5]York and Scarborough Teaching Hospitals NHS Foundation Trust, York, UK
[6]Department of Psychiatry, University of York, York, UK

**Acknowledgements** Creating Learning Environments for Compassionate Care (CLECC) 2020 University of Southampton. Used under non-exclusive licence.

**Contributors** MP and MJ led study conceptualisation and design, with contributions from CJ, JB and NS. MP and MJ led development of analysis plans, with contributions from CJ, CH and MT. MP led the writing process and drafted the original protocol with input from GJ and MJ. Critical review of the protocol and contributions to refinement to: codesign work package from GJ, MO, IF and MT; feasibility work package from GJ, CJ, JB, IF, CH, MO, KS, NS, MT and MJ; process evaluation work package from GJ, CJ, IF, MT and MJ. All authors took responsibility for the protocol and approved the final version of this paper.

**Funding** This work is supported by Yorkshire Cancer Research (Award reference number HEND405DEL).

**Competing interests** None declared.

**Patient and public involvement** Patients and/or the public were involved in the design, or conduct, or reporting, or dissemination plans of this research. Refer to the Methods section for further details.

**Patient consent for publication** Not applicable.

**Provenance and peer review** Not commissioned; externally peer reviewed.

**ORCID iDs**
Mark Pearson http://orcid.org/0000-0001-7628-7421
Catriona Jackson http://orcid.org/0000-0002-7806-7649
Jason Boland http://orcid.org/0000-0001-5272-3057
Najma Siddiqi http://orcid.org/0000-0003-1794-2152
Maureen Twiddy http://orcid.org/0000-0002-3794-1598
Miriam Johnson http://orcid.org/0000-0001-6204-9158

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
