## [Reviewer comments · BMJ Open]

ARTICLE DETAILS

TITLE (PROVISIONAL)	Improving the Detection, Assessment, Management, and Prevention of Delirium in Hospices (the DAMPen-D study): protocol for a co-design and feasibility study of a flexible and scalable implementation strategy to deliver guideline-adherent delirium care
AUTHORS	Pearson, Mark; Jackson, Gillian; Jackson, Catriona; Boland, Jason; Featherstone, Imogen; Huang, Chao; Ogden, Margaret; Sartain, Kathryn; Siddiqi, Najma; Twiddy, Maureen; Johnson, Miriam

VERSION 1 – REVIEW

REVIEWER	Mumford, Virginia Macquarie University, Australian Institute of Health Innovation
REVIEW RETURNED	24-Jan-2022

GENERAL COMMENTS	Thank you for asking me to review this protocol for a feasibility study to implement a CLECC intervention to assess adherence to delirium guidelines in a hospice setting. I have a few minor queries: P8 L13. It appears there is a reference missing P10 Is there a minimum length of stay for participants? P13 L28 to collect meaningful costing data, the team might also consider collecting the pay-grade of staff involved and volunteer time
---

REVIEWER	McKenzie, Cathrine King's College Hospital NHS Foundation Trust, Pharmacy
REVIEW RETURNED	06-Mar-2022

GENERAL COMMENTS	I have read this manuscript about 2 or 3 times now, to get an overview of what the investigators are proposing, Figure 1 does go some way to increasing my understanding of what the investigators are trying to achieve, but nevertheless it is still very complicated and a difficult read. Are all the lead clinicians at each hospice a core part of the investigator team ? The authors are proposing 3 different study designs in 3 hospices and an end report. To inform a future quasi-experimental multi-site comparative evaluation I suggest tables 1 and 2 be switched to better enable understanding, Article summary There are no limitations in the summary, please consider adding Introduction Explain why delirium screening is important in the hospice
---

	environment. Is there any data on the burden of delirium in the dying on healthcare professionals ? If there is please add it here. Methods Page 6, line 38, reduce the detail of the settings. Patient and public involvement Page 6, line 43, too long and detailed, summarise PPI section. Page 8, line 18 and line delete considered, delete (sometimes very different), line 20 delete 'open up new ways of thinking' perhaps to 'give way to different process' Page 8 line 35. One sentence describing diversity challenges is sufficient. Page 8 line 44 sentence that opens 'it could be the sharing of a personal or professional experience of delirium, delete text after workshop participant. Too much details. Line 46 merge end of sentence to 'how CLECC can be adapted for hospices and support implementation of delirium guidelines' Table 3 Is too text heavy and difficult to decipher, the authors could consider shortening the content somewhat by giving examples of what the discussion will involve rather than every detail. Page 10. Feasibility Are the study team suggesting the clinical leads take ownership of the 'CLECC-PAL' as well as running a hospice ? I am not convinced this will actually occur as the study team desires, as it seems very complicated. Page 11 line 55 add 'documented ' to inpatient delirium episodes. Table 4- add purpose to medication review , detect and minimise deliriogenic medication Page 12 line 33 remove justification for process evaluation and begin paragraph with 'We shall use realist evaluation.' . Line 40 delete 'used in this work package' Table 5- is this necessary Page 13 Page 13 line 32 delete all different type of staff and replace with healthcare staff including (then one or two staff groups). Page 13 line 52 over-complex sentence please simplify Page 14 line 20- delete re-read Page 15 Table 6 far too much text, simplify and summarise. Page 17-Progression to an evaluative study design Re-explaining the reasoning behind the study design (why the investigators did not select a randomised stepped wedge design has been expressed earlier in the paper) and I don't believe add to this section.
--	--

VERSION 1 – AUTHOR RESPONSE

Reviewer 1		
P8 L13. It appears there is a reference missing	The 'error reference not found' was a hyperlink to Figure 1 (Study flowchart and timeline summary) - this has now been corrected.	8
P10 Is there a minimum length of stay for participants?	There is no minimum length of stay for participants - as stated in the protocol, the baseline sample is of '50 consecutive patients who completed their in-patient stay immediately prior to the start of the hospice using CLECC-Pal' and the follow-up sample is of '50 consecutive patients completing their in-patient stay from week 4 of starting use of CLECC-Pal'.	-
P13 L28 to collect meaningful costing data, the team might also consider collecting the pay-grade of staff involved and volunteer time	Good point - we've added this.	12
Reviewer 2		
I have read this manuscript about 2 or 3 times now, to get an overview of what the investigators are proposing, Figure 1 does go some way to increasing my understanding of what the investigators are trying to achieve, but nevertheless it is still very complicated and a difficult read.	We have endeavoured to present each work package as clearly as possible, explicitly linking work packages to each of the aims and objectives, and cross-referencing between the work packages to show how they develop into more than the sum of their parts. We have also presented an overview of the study in a table (Table 1) and graphically (Figure 1). We recognise that the protocol remains quite complex, but are also strongly of the view that a protocol for research on a complex topic shouldn't be over-simplified , particularly if this would misrepresent the planned study (e.g. presenting the work packages in a straightforward linear way when the intention is for them to work iteratively and interactively). <="" span="" style="font-family:Arial; -aw-import:spaces">	-
Are all the lead clinicians at each hospice a core part of the investigator team ?	No, but as stated in Work Package 2, the study team will support lead clinicians to take ownership of, and implement, CLECC-Pal. We recognise that this rests on developing supportive working relationships with lead clinicians, which we are prepared to invest time in and which we believe will be facilitated by the co-design process in Work Package 1.	-
The authors are proposing 3 different study designs in 3 hospices and an end report. To inform a future quasi-experimental multi-site comparative evaluation. I suggest tables 1 and 2 be switched to better enable understanding,	The numbering of the tables reflects the order in which they are referred to in the text. As neither reviewer or editor has suggested we should re-structure the text, our view is that switching the order of these tables would be unlikely to better enable understanding. We are happy to consider further editorial direction about this.	-
Article summary There are no limitations in the summary, please consider adding	Done - please see response to Editor's comments above.	2
Introduction Explain why delirium screening is important in the hospice environment.	We acknowledge that we have used a number of terms in the Introduction to refer to areas that include hospices, e.g. 'adult palliative care settings', 'specialist palliative care units', 'palliative care teams' - this reflects the complex nature of palliative care delivery (delivered in different settings by teams of different composition), and we did not want to misrepresent cited sources by suggesting that findings	3

	related only to hospice care. We have therefore added a statement in the first paragraph noting that 'hospices are an important but under-researched setting for the delivery of delirium care'	
Is there any data on the burden of delirium in the dying on healthcare professionals ? If there is please add it here.	Brajtman et al citation added.	3
Methods Page 6, line 38, reduce the detail of the settings.	Understanding the impact of contextual differences between the settings is an important part of the study. As we state in the Introduction, 'guideline implementation requires a relevant and flexible strategy based on an understanding of how adaptation for different settings can be attained whilst retaining effectiveness.' We therefore do not think it is appropriate to reduce the detail (two short sentences) of the settings.	-
Patient and public involvement Page 6, line 43, too long and detailed, summarise PPI section.	The PPI section is three sentences long and prefer not to reduce the detail as it is consistent, in our view, with BMJ Open's author guidelines and with the level of detail in other protocols published recently in BMJ Open. We are happy to consider editorial direction regarding this.	-
Page 8, line 18 and line delete considered, delete (sometimes very different), line 20 delete' open up new ways of thinking' perhaps to 'give way to different process'	'(sometimes very different)' deleted, but we have kept 'open up new ways of thinking' as 'give way to different process' would not be consistent with the cited source (Iedema et al 2010)	8
Page 8 line 35. One sentence describing diversity challenges is sufficient.	The second sentence (11 words) refers to how we shall work with our PPI collaborator to monitor diversity of participants in the workshops. We consider this to be a vital role and would therefore prefer to keep the sentence.	-
Page 8 line 44 sentence that opens 'it could be the sharing of a personal or professional experience of delirium, delete text after workshop participant. Too much details.	These examples of 'touch points' that could be used in the co-design workshops are important to state, as it would be incorrect to state that these would only be 'sharing of a personal or professional experience by a participant'. We have therefore not truncated the text.	-
Line 46 merge end of sentence to 'how CLECC can be adapted for hospices and support implementation of delirium guidelines'	Done	8
Table 3 Is too text heavy and difficult to decipher, the authors could consider shortening the content somewhat be giving examples of what the discussion with involve rather than every detail.	We believe it is important to retain this detail as co-design processes are often poorly-reported. It is unclear to us how providing examples would enable the text to be shortened considerably. We are happy to consider editorial views on this.	-
Page 10. Feasibility Are the study team suggesting the clinical leads take ownership of the 'CLECC-PAL' as well as running a hospice ? I am not convinced this will actually occur as the study team desires, as it seems very complicated.	As a real-world feasibility study, we are seeking to understand how hospices can 'take ownership' of an intervention such as CLECC-Pal, as future evaluation of an intervention that is possible to deliver in routine practice will be key. We recognise that enabling clinical leads to 'take ownership' is neither straightforward nor guaranteed, but (as stated in our earlier response), this rests on developing strong working relationships with lead clinicians, We are prepared to invest time in this (and have been doing so to-date) and the emerging signs	-

	are that this is being facilitated by the co-design process in Work Package 1.	
Page 11 line 55 add 'documented ' to inpatient delirium episodes.	Done	11
Table 4- add purpose to medication review , detect and minimise deliriogenic medication	Done	11
Page 12 line 33 remove justification for process evaluation and begin paragraph with 'We shall use realist evaluation.'	Although the role of process evaluations is now more widely-accepted, our view is that it is nevertheless important to acknowledge recent work and in particular their explicit recognition in the recently-revised MRC Complex Interventions Framework (we have added a citation to this).	12
Line 40 delete 'used in this work package'	Done	12
Table 5- is this necessary	Yes, the table defining realist terms is necessary - the importance of doing this has been identified by peer-reviewers of every other realist study we have conducted.	-
Page 13 line 32 delete all different type of staff and replace with healthcare staff including (then one or two staff groups).	Done	13
Page 13 line52 over-complex sentence please simplify	Split into two sentences	13
Page 14 line 20- delete re-read	'Re-reading' is a vital part of the familiarisation process in qualitative analysis, so we have retained this.	-
Page 15 Table 6 far too much text, simplify and summarise.	In our view it is important to clearly link the interview questions to the underlying theory, which Table 6 does. 'Summarising' risks breaking the link between the two. Nevertheless, we appreciate that this is a large and text-heavy table and would appreciate editorial direction about whether it would be preferable for the Table to instead be an online supplemental file.	-
Page 17-Progression to an evaluative study design Re-explaining the reasoning behind the study design (why the investigators did not select a randomised stepped wedge design has been expressed earlier in the paper) and I don't believe add to this section.	Although the 'reasoning behind [a randomised stepped wedge design]' was not 'expressed earlier in the protocol', we have revised the phrasing at the start of this section to avoid giving this impression.	17

VERSION 2 – REVIEW

REVIEWER	Mumford, Virginia Macquarie University, Australian Institute of Health Innovation
REVIEW RETURNED	22-Mar-2022
GENERAL COMMENTS	The authors describe a mixed method feasibility study to investigate an implementation tool to assess compliance with delirium - the authors have identified a gap in application of these guidelines in a known high risk patient group. I did have some questions about the patient record collection. P10 L 9 – it is not clear what the delirium day is – is this the data

	collection day across the hospices - or the total number of patient days with delirium (see P11 L43 below)? P10 L36 the authors could clarify whether the post collection excludes patients from the baseline review - as this could confound the results P10 L58 – Multiple records of the same patient may need to be treated as clusters in the analysis P11 L43 – This concept of "delirium day" needs clarifying- P11 L33 It would be helpful to discuss the number of expected delirium episodes given only 50 baseline and 50 follow-up patients will be evaluated. What is the main effect being investigated here # patients with delirium, # patient days with delirium, # patients assessed for delirium, or other metrics from the guidelines? and will the projected number of episodes be enough to measure a clinical and statistical difference. As the team have already done some preliminary work in this area, some details would be helpful Other points – is underlying dementia identified? Does the study include both incident and prevalent delirium? I also had a question as to whether the journal accepts protocols submitted after data collection has started.
--	---

VERSION 2 – AUTHOR RESPONSE

Reviewer 1 comments:

a) P10 L 9 – it is not clear what the delirium day is – is this the data collection day across the hospices - or the total number of patient days with delirium (see P11 L43 below)? / P11 L43 – This concept of "delirium day" needs clarifying-

>>> Definition now included where delirium days are first mentioned in the manuscript (p4) pages 10 and 11 ('a delirium day being one where the patient was classed as having delirium using Inouye et al's chart-based instrument').

b) P10 L36 the authors could clarify whether the post collection excludes patients from the baseline review - as this could confound the results

>>> Due to the requirements of the Health Research Authority's Confidentiality Advisory Group, we are not permitted to retain any non-anonymised data that would permit identification of individuals on a subsequent admission, therefore we are unable to exclude patients about whom data was collected in the 'pre-' stage. We are also unconvinced that this would be the right thing to do scientifically as the study aims to collect data about the impact of changes in practice (through the CLECC-Pal intervention) on delirium outcomes – hospice in-patients are not on a linear course to recovery, so it is equally valid to assess the 'post-' impact whether or not a patient was included at the 'pre-' stage.

c) P10 L58 – Multiple records of the same patient may need to be treated as clusters in the analysis

>>> We state in the manuscript that 'Where a person experiences multiple episodes of delirium within one admission, each episode will be recorded separately and linked through the anonymised case

number' (p10, L58), which would enable appropriate treatment in the analysis. Due to the requirements of the Health Research Authority's Confidentiality Advisory Group, we are not permitted to retain any non-anonymised data that would permit identification of individuals on a subsequent admission.

d) P11 L33 It would be helpful to discuss the number of expected delirium episodes given only 50 baseline and 50 follow-up patients will be evaluated. What is the main effect being investigated here # patients with delirium, # patient days with delirium, # patients assessed for delirium, or other metrics from the guidelines? and will the projected number of episodes be enough to measure a clinical and statistical difference. As the team have already done some preliminary work in this area, some details would be helpful

>>> These questions all relate to Work Package 2 (Feasibility Study), for which it's worth keeping in mind the study's objectives in relation to this Work Package. Our objectives do not relate to effectiveness, but to demonstrate if it is possible to:

- Systematically and reliably collect data (including delirium diagnosis) from clinical records in a way that minimises burden for patients, families, and staff.
- Collect measures of staff engagement with the implementation strategy, delivery of guideline-adherent delirium care, and the costs of staff involvement.

Taking each of the points in turn:

1. Number of expected delirium episodes – relevant results of our pilot work are now described in relation to the sample size (p11).

2. Main effect being investigated – As stated in the manuscript (p10, L47-50), the main effect being investigated is the impact on delirium days – subsequent to point a), we have added the definition ('a delirium day being one where the patient was classed as having delirium using Inouye et al's chart-based instrument') earlier in the manuscript where delirium days are first mentioned (p4). It is also important to consider this primary outcome in relation to other data collected from case records that will enable us to link the extent of guideline implementation to impacts on delirium days, namely: 'The instrument (data extraction pro-forma, see online supplemental file 1) will enable us to assess whether case-note recorded symptoms of delirium can be linked to time-points during the person's admission when actions around delirium assessment, management and prevention (consistent with guidelines) did or did not take place. Our 'expanded' version of the instrument will include questions about other actions to support delirium assessment, management and prevention that may be recorded in the notes' (p10, L50-57)

3. Will the projected number of episodes be enough to measure a clinical and statistical difference? – Again, as this is a Feasibility Study to demonstrate that data about delirium outcomes and staff actions can be systematically and reliably extracted from clinical records, we have not formally powered the study but do state how we shall use the data collected to calculate sample size for a future multi-site evaluative study (p18, L6-23). On the basis of Watt et al's (2019) systematic review of incidence and prevalence of delirium across palliative care settings (Ref.8), which reported that one-third of people in adult palliative care settings had delirium on admission, with two-thirds developing delirium during the admission, it is reasonable to expect that we will detect a substantial number of delirium days, but detecting a change in this outcome is not one of our objectives.

e) Other points – is underlying dementia identified? Does the study include both incident and prevalent delirium?

>>> The Inouye et al tool we are using would not identify undiagnosed dementia, but we are extracting data about known dementia diagnosis. We will be assessing the medical record for the

whole inpatient stay so that incident and prevalent delirium can be measured and distinguished.

f) I also had a question as to whether the journal accepts protocols submitted after data collection has started

>>> We understand from the journal editors that protocols may be accepted after data collection has started but not if final data analysis has commenced. Our study is at the stage of data collection, not analysis.

VERSION 3 – REVIEW

REVIEWER	Mumford, Virginia Macquarie University, Australian Institute of Health Innovation
REVIEW RETURNED	20-Apr-2022
GENERAL COMMENTS	Thank you clarifying the points raised in the initial review.